# Data Glove with Self-Compensation Mechanism Based on High-Sensitive Elastic Fiber-Optic Sensor

**DOI:** 10.3390/polym15010100

**Published:** 2022-12-26

**Authors:** Hui Yu, Daifu Zheng, Yun Liu, Shimeng Chen, Xiaona Wang, Wei Peng

**Affiliations:** 1School of Physics, Dalian University of Technology, Dalian116024, China; 2School of Optoelectronic Engineering and Instrumentation Science, Dalian University of Technology, Dalian 116024, China; 3Marine Engineering College, Dalian Maritime University, Dalian 116026, China

**Keywords:** elastic fiber, data glove, self-compensation, flexible sensor, human-computer interaction

## Abstract

With the development of virtual reality (VR) interaction technology, data glove has become one of the most popular devices for human-computer interaction. It’s valuable to design high-sensitive and flexible sensor for data glove. Therefore, a low-cost data glove based on self-compensating elastic optical fiber sensor with self-calibration function is proposed. The tunable and stretchable elastic fiber was fabricated by a simple, economical and controllable method. The fiber has good flexibility and high stability under stretching, bending and indentation deformation. The optical fibers are installed in the sensor in a U shape with a bending radius of 5 mm. Compared with the straight fiber, the response sensitivity of the U-shaped fiber to deformation is increased by about 7 times at most. The reference optical fiber is connected to the sensor, which effectively improves the stability and accuracy of the sensor system. In addition, the sensors are easy to install so that the data gloves can be customized for different hand shapes. In the gesture capture test, it can respond quickly and guide the manipulator to track the gesture. This responsive and stable data glove has broad development potential in motion monitoring, telemedicine and human-computer interaction.

## 1. Introduction

Recently, with the development of virtual reality (VR) interaction, researchers are increasingly keen on flexible, stretchable and wearable sensor devices that can track complex human movements [1,2,3,4,5,6,7,8,9]. Since the human hand has more than 20 degrees of freedom, it has flexible functions in communication and operation [10] and can transmit a large number of information. It is one of the body’s most important organs for communicating with the outside world. Data gloves have become one of the most popular devices of human-computer interaction, widely used in medical, education, games and other fields [11,12,13,14,15].

Currently, some data glove devices based on point tracking and recognition technology [16,17], computer vision technology [18,19], FBG sensor technology [20,21] and inertial sensor technology [22,23] have been proposed. However, data gloves based on the above technologies have different disadvantages. In point tracking and recognition technology, optical, acoustic and electromagnetic marker points are mainly used. However, optical markers are easily obscured during hand movements. Acoustic and electromagnetic marker point technology also have certain limitations, such as susceptibility to electromagnetic interference and low resolution. Data gloves based on computer vision technology need to work in specific system operating environment and ambient light conditions. FBG sensors have high sensitivity and accuracy. However, the complexity of measurement system restricts its development in the field of data glove. The data glove based on inertial sensor has fast response in gesture capture. However, inertial sensors are not suitable for working for a long time due to the accumulated measurement error. In addition, to fully capture the movements and senses of the hand, data gloves based on the above technologies would need to have multiple sensors embedded in each finger. This fusion of sensors increases the complexity of the data glove system and reduces its portability and utility.

In addition to the traditional technologies mentioned above, the application of elastic fiber optic sensors in data gloves has attracted extensive attention because of their advantages such as stretchable, small size, high sensitivity, fast response and anti-electromagnetic interference [24,25]. So far, elastic optical fiber sensors have been widely used in various practical applications due to the low-cost, scalable, simple and diverse production methods of elastic material fibers [26,27,28,29,30]. Among them, Leber et al. developed a thermo-plastic optical fiber sensor for detecting extreme deformations through the wavelength-dependent changes in light transmission [31]. The fabricated fibers were able to reversibly maintain strains of up to 300% while guiding light. Yang et al. reported a sensor based on dye-doped PDMS fiber [32], which has good durability, reliability and long-term stability. The tensile strain can be measured quantitatively according to the change of light absorption through dye-doped fiber. The sensor has a linear and repeatable response over a wide dynamic range of up to 100%. In 2019, Sheng et al. reported a graphene-supplemented PDMS fiber [33]. The fiber has excellent strain sensing performance, high sensitivity, tensile range up to 150%. In 2022, Gan et al. reported a stretchable optical sensor with strain decoupling capability [34]. The stretchable fiber is made of fluorescent nanoparticles and silicone-based elastomers, which can achieve efficient excitation light transmission and fluorescence collection. These studies have expanded the application of optical sensors such as elastic fiber sensors in the field of wearable devices. However, optical sources used in fiber optic sensors are often disturbed by factors such as current fluctuations, which can not be ignored. This is especially true for mobile devices with their own batteries. As shown in Figure 1a, the aging of the battery reduces the current to the optical source, which leads to the change of light intensity and brings errors to the measurement system. Therefore, for further meet the requirements of wearable devices for sensor stability and reliability, it’s crucial to take effective and easy to implement measures to reduce the system error caused by the unstable optical source. Furthermore, low-cost and tractable flexible sensors with fast response, high sensitivity and super-stability are very necessary for the construction of user-friendly intelligent wearable devices.

Hereby, a low-cost data glove based on a self-compensating elastic optical fiber sensor with self-calibration function is proposed for gesture capture. The tunable and stretchable elastic fiber was fabricated by a simple, economical and controllable method. Figure 1b shows that the elastic optical fiber has good flexibility and stability. It can not only maintain outstanding sensing properties at 10 °C to 50 °C, but also exhibit high stability after deformation such as stretching, bending and indentation. An additional communication-grade plastic fiber (attenuation less than 180 dB/km) is connected to the sensor as a reference signal (Figure 1c). The reference fiber can effectively reduce the error of the sensor system. The structure diagram of the data glove is shown in Figure 1d. The optical fibers are installed in the sensor in a U shape with a bending radius of 5 mm. Compared with the straight fiber, the response sensitivity of the U-shaped fiber to deformation is increased by about 7 times at most. In addition, the sensors are easy to install so that the data gloves can be customized for different hand shapes. To sum up, the production process of the data glove is simple, the sensor component is universal, and the low-cost glove enables real-time monitoring of finger movements(the actual price of each component is recorded in Appendix A). This will facilitate the development of data gloves in areas such as motion monitoring, telemedicine and human-computer interaction.

## 2. Materials and Methods

### 2.1. Materials and Instrumentation

Double additive liquid silicone rubber, double component Room Temperature Vulca-nized (RTV) silicone, demoulding agent, acrylic hollow tube, heat shrinkable tube and a number of positioning plugs, vacuum pressure pump (Yancheng ricky labora-tory Instrument Co., Ltd., Yancheng, China, 10 L/min), syringes (Weihai Wego Medical Systems Co., Ltd., Weihai, China, 10 mL) and electric blast drying oven (Tianjin Hongnuo instrument Co., Ltd., Tianjin, China, 101-00BS). The stepper motor (Beijing Times Chaoqun Electrical Technology Co., Ltd., Beijing, China, CBX1605-150) was used to generate tensile strain for the flexible optical fiber. The sliding table (Taihe Qicheng Hardware & Building Materials Co., Ltd., Taihe, China, RS60-L) was used to apply bending and torsional strains on the flexible optical fiber. The tension meter (Wenzhou Wanda Electronics Co., Ltd., Wenzhou, China, WDF-100) was used to apply side force on the flexible optical fiber.

### 2.2. Fabrication of Elastic Optical Fiber

Bending, stretching and other deforms lead to changes in the propagation path of light in the fiber, which cannot meet the conditions of total internal reflection. The radiation mode in the fiber replaces the transmission mode, so that a part of the light is scattered through the cladding, resulting in loss. The natural movement of the hand is a regular movement with internal constraints: the movement of the three phalanx bones of the fingers is in the same plane, and the muscle and skin and other soft tissues of the hand make the movement of one knuckle must cause the joint movement of adjacent knuckles. In this case, the degree of finger bending can be determined by measuring the total loss of the whole finger.

The elastic fiber in this work is a double-layer coaxial cylindrical rubber structure. The core layer and cladding layer were made of silicone rubber with different refractive index (Core: room temperature vulcanized (RTV) silicone rubber, RI = 1.413; Cladding: two-component liquid silicone rubber, RI = 1.411). The cross section diameters of the fiber core and the whole fiber are 1.5 mm and 2.9 mm respectively, and the length of the fiber is 40 cm. The manufacturing process of elastic fibers is shown in Figure 2a. The process of making the cladding first and then filling the core into the cladding is adopted. This way can not only reduce the interference of external debris in the production process, but also simplify the production process.

The solvent and crosslinking agent are mixed according to the weight ratio of 10:1. After stirring fully, the liquid is put into the vacuum pressure pump for foaming. Inject the defoaming liquid into syringes A and B and let it stand at room temperature until there are no bubbles in the liquid (bubbles can affect the refractive index and toughness of the fiber). To facilitate release after curing, we spray oil release agent evenly on the inside of the outer mold (inner diameter: 2.9 mm acrylic hollow tube) and the outside of the inner mold (inner diameter: 1.5 mm heat shrinkable tube) and the surface of the fixed plug. After 20 s, place the inner mold in the outer mold and secure it with a fixed plug to ensure that the inner mold is centered in the outer mold. The defoaming cladding solution was injected into the interlayer between the outer mold and the inner mold with syringe A and placed at room temperature until the liquid was completely cured. Then the drying oven is heated to 100 °C, and the inner mold is pulled out when it shrinks to 1/4 of its diameter. The resulting stretchable transparent hollow tube is the cladding of the elastic fiber. The solution in syringe B is injected into a hollow tube, which is prone to tiny bubbles. So let it stand until the air bubbles are drained. When the bubbles are completely drained, the fibers were placed in a drying oven at 80 °C to complete curing. Then the elastic fiber was pulled out of the outer mold. Figure 2b is the elastic fiber after fabrication, which can be stretched and bent at will and has certain toughness and recoverability.

### 2.3. Installation of the Elastic Fiber Sensors into Data Glove

There are many uncertainties in the use of data gloves, such as the effect of the cartilage tissue of the hand on the sensor. Elastic fiber is a kind of force sensitive material in nature, and even the slightest interference will affect the transmission of light in it. Therefore, it is necessary to fit the user’s hand as closely as possible when installing the elastic fiber.

As shown in Figure 1d, a U-shaped elastic fiber sensing structure is designed. Compared with the straight fiber, the U-shaped structure effectively increases the attachment point of the fiber on the glove surface, which enhances the sensing ability of the data glove to very small deformations. The elastic fiber twice the length of a finger was folded in half through a 5 mm shaft wheel. Instead of mounting the fiber directly on the glove, it is placed in a latex tube. The latex tube is then glued to the glove. Such a design can not only increase the deformation area of the elastic fiber, disperse the pressure at the knuckle, and increase the life of the fiber, but also prevent the relative displacement in each use under the condition of fitting the user’s finger. The U-shaped design amplifies the tiny bends of the finger during data acquisition, increasing the sensitivity of the data glove. To enhance the bonding strength between the latex tube and the glove, the latex tube was treated with surface treatment agent before bonding, and then bonded with silica gel special soft glue, which not only ensured the softness of the glove, but also increased the accuracy of the sensor to collect finger movement, and prevented some unnecessary interference.

The elastic fiber sensors quantify the deformation of the fiber by measuring the total optical loss of the entire finger during propagation. In order to ensure the portability of the data glove, we choose 5 mm ultra-bright light-emitting diode wire lamp beads as the optical source, two button batteries provide power for it, and an 8-megapixel micro camera is used to collect the gray value changes of the fiber section. An additional 5 cm communication-grade plastic fiber (attenuation less than 180 dB/km) is installed between the optical source and the camera to compensate for changes in light intensity caused by the gradual decline of battery power during use. The entire acquisition module is encapsulated in black acrylic sheets to prevent interference from external light and lamp pearlescent sources. Epoxy putty is used for the internal fixed camera and elastic optical fiber to ensure that the data glove can still collect data stably during use. The elastic fiber optic sensor can be adapted to different types of textile gloves because its components are easy to install, which allows our data gloves to be customized for different hand shapes.

## 3. Results and Discussion

### 3.1. Properties Characterization of Elastic Fiber

To examine the mechanical and optical properties of the fibers, tensile, bending and indentation tests were performed. And the results were analyzed by normalized spectral analysis. The optical fiber is mounted on the data glove in a U-shaped bending state as shown in Figure 3a. It is important to select the appropriate bending radius to avoid extinction to meet the sensor requirements. Therefore, the normalized spectrum at different bending radii were measured. Figure 3b shows that the light passing through the elastic fiber decreases as the bending radius decreases. As the deformation sensor uses 5 mm ultra-bright LED wire red lamp bead as the optical source, we choose the bending radius-normalized light intensity relation diagram at 650 nm (Figure 3c). It can be seen that at least 30% of the light can still pass through the fiber when the bending radius of the fiber is reduced from infinity to 2.5 mm, which is a necessary condition for the fiber to be used for sensing. Based on the results of the bending test and the design of the data glove, we decided to install the optical fiber on the data glove with a bending radius of 5 mm. Therefore, to evaluate the mechanical properties of the fiber under bending condition, the strain-normalized light intensity curve of the fiber under folding condition was measured at 650 nm. As shown in Figure 3d, the mechanical integrity of the fiber can still be maintained under 100% tensile strain. When the strain is between 100% and 110%, the extinction point A appears in the strain normalized light intensity curve of the fiber. The point A indicates that the fiber extinction occurs when the elongation is greater than 100%, but there is no fracture phenomenon.

To characterize the optical properties of the fiber, the fiber was successively shortened from 20 cm to 10 cm and the normalized light intensity was measured to quantify the propagation loss(the relevant measurement methods are shown in Appendix A). In Figure 3e, the normalized light intensity decreases with increasing fiber length, which is due to the longer path of light through the attenuating medium due to increasing fiber length. In the indentation test, a pressure of 10 to 100 N was applied sequentially on the side of the fiber at 10 N intervals, and the pressure-normalized light intensity curves were obtained (Figure 3f). Figure 3f shows that the light passing through the fiber decreases as the pressure increases. In order to more intuitively express the relationship between optical loss in the fiber and the pressure on the fiber, the pressure-normalized light intensity relationship diagram at 650 nm was selected, as shown in Figure 3g. The transmission loss of optical fiber caused by side pressure is caused by the fact that the propagation path of light does not satisfy the condition of total internal reflection. However, the results show that our elastic fiber can still pass through about 30% of the light when subjected to 50 N pressure. This indicates that the fiber used for sensing is highly resistant to lateral pressure. In addition, the deformation sensor mounted on the data glove will hardly be subjected to more than 30 N pressure in practical applications. Even so, during the use of the data glove, the optical fiber should be protected from side pressure as much as possible.

After examining the mechanical and optical performance of the elastic fibers, we evaluated the optical response of the fibers to perturbations caused by repeated bending and stretching. To test the bending response, we bent the fiber from the straight state to the bending radius of 5 mm, and then returned to the straight state as a cycle as shown in Appendix A. After 100 cycles, the normalized spectrum of the fiber was measured after each cycle, as shown in Figure 4a. During 100 cycles of bending, the normalized intensity distribution in the fiber remains above 0.98, which means that the loss is only 2%. In the tensile cycle test, the tensile strain from 0% to 50% and then back to 0% was defined as one cycle. As seen in Appendix A, a fiber with a bending radius of 5 mm was stretched repeatedly over 100 cycles. And the normalized spectrum of the fiber was measured after each cycle, as shown in Figure 4b. During 100 cycles of stretching, the normalized intensity distribution in the fiber remains above 0.94. The light intensity drops directly from 1.00 to 0.98 after the first cycle. This phenomenon is attributed to the Mullins effect. The Mullins effect describes the phenomenon of stress softening and hysteresis in elastic materials [35]. Therefore, to eliminate the Mullins effect, each fiber was pretreated with at least 100 cycles of stretching before being attached to the sensor. In fact, the strain applied during pretreatment should be higher than the maximum strain that can be encountered during the sensor’s use. The above two experiments indicate that the fiber is highly stable after bending and stretching deformation, which ensures the reproducibility of the experimental results.

To examine the influence of U-shaped structure on sensor performance, relevant tests were conducted to compare the sensing performance of the 10 cm straight fiber and the 10 cm U-shaped fiber. We applied the tensile strain from 0% to 50% and then back to 0% to the fibers of the two forms at the interval of 10% elongation, and measured the relationship between the light intensity and the strain in the fibers of the two forms. It can be seen from Figure 4c, in the process of stretching, the light in the elastic fiber will change simultaneously with the change of tensile strain, which is a necessary condition for the deformation sensor to be able to complete the sensing task. In addition, the sensitivity of light response to strain in the U-shaped fiber is up to 7 times higher than that in the straight fiber. This indicates that the fiber in the bent state has a higher strain response sensitivity, which is the most important reason to install the fiber in the bent state on the glove. In order to evaluate the influence of tensile strain on optical properties of U-shaped fiber, the U-shaped fiber was stretched for 50 cycles at 10–50 °C above zero within the strain range of 0%~50% at 10 °C intervals, and the normalized spectrum of the fiber after each cycle was measured (Figure 4d–h). The cycling process at 10 °C is shown in Figure 4d. During 50 cycles of stretching, the normalized light intensity in the fiber is kept between 0.95 and 1.00, and only 5% of the normalized light intensity is lost. As can be seen in Figure 4e, when the temperature rises to 20 °C, the light in the fiber loses at most 10% after 50 cycles of stretching. Similarly, Figure 4f–h show that the light loss in the fiber does not exceed 15% at most during the cycle at temperatures of 30 °C, 40 °C, and 50 °C. To compare the results at different temperatures more intuitively, the average normalized light intensity at 650 nm was selected at 5 temperatures, as shown in Figure 4i. Obviously, the loss after the cycle will gradually increase with the rise of the ambient temperature. However, at temperatures below 50 °C, the light loss does not exceed 15%. As a result, our elastic fiber still exhibits extremely high stability in the bent state when operating at temperatures ranging from 0 to 50 °C.

In summary, temperature variation, bending, stretching, indentation and other deformation may cause certain optical loss. When the temperature is lower than 50 °C, the loss caused by stretching is small, and the fiber shows high sensing stability. In addition, elastic fibers can maintain their sensing properties after bending and indentation deformation. The elastic fiber can not only exhibit high stability, but also ensure that the optical signal changes with the change of shape variables in the deformation process, which is the basis of realizing optical fiber sensing. Most importantly, compared with the straight fiber, the response sensitivity of the U-shaped fiber to deformation is increased by about 7 times at most.

### 3.2. Data Collection and Processing

In order to more clearly observe the process of quantifying finger deformation with data glove, Raspberry Pi (RPi) manipulator (6 degrees of freedom) is selected as the control and gesture tracking display platform. In the RPi environment, Python is used to process the gray value information in the data glove. The whole process can be divided into two modules: data collection and data processing. The processed information is transmitted to the RPi system in real time, and then the manipulator displays the gesture tracking effect of the data glove. It can be seen from Figure 5, the data collection and processing process includes initialize settings and data processing.

#### 3.2.1. Initialize Settings

Since each person’s hand shape and the initial value of the optical source are various, the initial state of wearing gloves may not be consistent each time. Therefore, it’s essential to adjust the zero position and amplitude of the sensor when using the data glove. During the adjustment process, a series of specific gestures need to be made in advance to obtain a value that matches the shape of the hand, so as to obtain a mild mapping relationship between the manipulator steering gear and the data glove, which is the premise of data processing.

#### 3.2.2. Data Processing

##### Fine-Tuning for Robustness

In data acquisition and glove data processing, the response speed, accuracy and robustness of the control system are the key factors to determine the system performance. Gesture capture requires the system to respond in time and accurately transmit data to the manipulator, so an appropriate and effective algorithm is necessary. When encountering some abnormal situations, it is particularly important for the system to maintain stable robustness. It can be seen in Figure 6, in the process of using gloves, the mapping relationship was fine-tuned three times to avoid the mapping relationship deviation caused by the inaccurate maximum and minimum gray value and amplitude obtained in the pretreatment. In order to ensure the stability and robustness of the system (there are some abnormal situation, for example, the data glove is subjected to external force and the value of the sudden change), a series of judgments and settings are necessary before fine-tuning the mapping relationship. When the fluctuation of limit gray value exceeds 50%, the system will conduct statistics on the occurrence of such limit value. The adjustment can be made only if the limit value fluctuates within 5% for five times.

##### Principle of Self-Compensation

It is difficult to replace the battery in time during data glove use. So as the battery’s power dwindles, the current to the optical source decreases, causing it to emit less light. This directly reduces the light in the fiber, which leads to the acquisition of the cross-sectional gray value less than the theoretical value by the data acquisition camera. This will greatly increase the error of gesture capture experiments. To compensate for this error due to battery aging, an additional 5 cm communication-grade plastic fiber (attenuation less than 180 dB/km) was installed between the optical source and the data acquisition camera as a control. In gray image, brightness is equivalent to gray level. Therefore, we define the corrected gray scale (*G_c_*) and express it as follows:(1)$$G_{c}=\frac{L_{i}\left(I_{0}-I_{r}\right)}{I_{0}}+I_{i}\;\;(i=1,2,3,4,5)$$
where, *I*_0_ represents the initial luminance of the plastic reference fiber section, which is approximately equal to the initial luminance of the optical source. *I_r_* represents the real-time brightness of the plastic reference fiber section, which is approximately equal to the real-time brightness of the optical source. When *i* = 1,2,3,4,5, *L_i_* and *I_i_* represent the initial and real-time luminance of the fiber section corresponding to each finger, respectively. In the data glove design, five optical fibers for sensing and a plastic reference fiber are mounted between the same data acquisition camera and the same optical source. When the brightness of the optical source decreases during the application of the sensor, the cross-sectional brightness of the five sensing fibers decreases proportionally with the brightness of the optical source. This decrease of the same proportion can be eliminated by conversion through Equation (1), and the corrected gray value close to the theoretical value can be obtained. This makes up for the experimental error caused by battery aging, which is extremely important for the data glove to give users a good experience.

To verify the effectiveness of the self-compensation function, relevant tests were carried out to study the influence of current (the main factor affecting the stability of the optical source) on the sensor system. The current is regulated by increasing the number of resistors (750 Ω each) in the optical source circuit. The gray curve of three same optical sources without reference fiber was measured. Then, Equation (1) is used to process these three groups of data, and three modified gray curves are obtained. As can be seen from Figure 7a, when there is no reference fiber, the current fluctuation has great influence on the sensor system. By introducing the reference fiber into the sensor system, the experimental error caused by current fluctuation can be compensated. The test results show that the reference fiber can effectively improve the stability and accuracy of the system, which is consistent with the theoretical analysis results above.

##### Intelligent Prediction

Because the hand has muscles, skin and other soft tissues, the movement of one finger may affect the output of adjacent sensors that are not moving, causing the manipulator to make the wrong action. To solve this problem, we repeated the gesture capture test and found that it followed a pattern. For example, even though the ring and index fingers don’t move, their corresponding sensors respond to the bending of the middle finger. This phenomenon can be “predicted” before test, so intelligent prediction is added to the program to correct these errors. The intelligent prediction is divided into single finger movement and finger combination movement.

Single finger movement. Two kinds of tests were carried out, namely continuous movement of single finger and cyclic action of each finger, to collect the changes of corresponding gray values and form a set of data. Then the gray value changes of the implicated fingers were found and recorded. In the whole test, if the implicated cases reached more than 90% of the whole, they would be included in the compensation set.

Finger combination movement. It involves two fingers, three fingers and four fingers combined movement. Include all possible gestures, collect each gesture repeatedly, record the change of the corresponding gray value, and also find the implicated fingers. If the implicated cases reach more than 90% of the whole, they will be included in the compensation set.

In formal gesture tracking, when the data glove makes an action, the system will capture the finger that has changed significantly, and then it will search for the corresponding situation in the compensation set. If it is found, it indicates that there is finger involvement, and the gray value of the implicated finger will be compensated to some extent. After extensive testing and data analysis, a variety of similar situations are collected to form an action-related recognition library. According to these error patterns, the compensation function relationship can be obtained to reduce the misjudgment of fluctuation. In data processing, the fast response of data is extremely important to the system, which is directly related to the real-time tracking of hand gestures. In addition, the fast response of the sensor mainly depends on the collected frame rate of the camera. The 8-megapixel camera, with a dynamic frame rate of 25 fps, can collect data 18 times per second and contain 90 data points to ensure fast response of data.

### 3.3. Gesture Capture Tests

The interface of the data glove can be connected to the console of the manipulator, and the collected gray value is used as the password to guide the manipulator to complete gesture capture. The sensing performance of the data glove is tested in two states: static test and dynamic test. The static test is to observe whether the gray value of the fiber section changes accordingly when the data glove makes corresponding gestures, so as to understand the accuracy of the elastic fiber quantifying finger bending more clearly. The dynamic test evaluates the real-time performance and stability of the sensor by photographing the gesture tracking effect of the manipulator.

#### 3.3.1. Static Test

In the static test, six consecutive actions were captured to observe the variation of the gray value of the elastic fiber under different gestures. As shown in Figure 7b, with the change of gesture, the change of corresponding curve can be obviously observed. The bending of the finger causes the fiber to deform and thus cause radiation loss. As a result, the gray value of elastic fiber section decreases. When the finger is gradually extended, the gray value returns to the initial value. When holding a gesture constant, the gray value will also remain the same. Because the length of elastic fiber is different, and the packaging force of the packaging material is different to the elastic fiber, the gray value amplitude interval is different. But this difference is only reflected in the value of the maximum and minimum gray value. Each elastic fiber almost follows the same law in the overall trend of change. It is worth noting that when one finger is bent, the corresponding curve of the other fingers is likely to fluctuate, rather than a straight line as expected. It’s caused by the physiology of the hand. The motion of each knuckle is not independent. When making different gestures, there is a correlation between the fingers. However, the associated fluctuations are not significant enough to affect the use of data gloves.

To observe the process of elastic fiber quantization finger in detail, the index finger was selected as a representative, and six extension ranges were selected, from clenching-extension-clenching, and the extension range gradually increased until the complete extension. As can be seen in Figure 7c, with the stepwise increase of finger extension amplitude, the corresponding gray value also increases stepwise. In addition, during the use of the data glove, the elastic fiber will be repeatedly affected by the pressure caused by bending, so it is also important to verify whether the elastic fiber can maintain high stability in the glove. Therefore, in the process of repeatedly bending and extending the index finger for 50 times, the average gray value curves are shown in Figure 7d. After 50 cycles, the fluctuation error of gray value is not more than 5%, that is, it is kept in a stable interval.

#### 3.3.2. Dynamic Test

To evaluate the repeatability and reproducibility of the data glove, the static test actions were repeated with the data glove three cycles during the dynamic test. As shown in Appendix A, the data glove exhibits excellent repeatability and reproducibility in dynamic test. And the video screenshots of Appendix A are shown in Figure 8. During the dynamic process, the data glove can accurately identify and track the change of hand gestures in real time, and then guide the manipulator to make the correct action. The time delay of the data glove is about 0.4 s. In addition, increasing the data collection frequency can further reduce the delay. Therefore, our data glove can fully meet the real-time interaction with the manipulator. We believe it has a bright future in virtual reality interactions.

## 4. Conclusions

The proposed data glove based on self-compensating elastic fiber optic sensor with self-calibration function. The stretchable elastic silicone rubber fiber was manufactured by simple, stable, economical and controllable method. The light loss of the optical fiber is not more than 2% when the optical fiber is repeatedly bent 100 cycles with a bending radius of 5 mm. The U-shaped fiber has good repeatability and mechanical stability in 0~100% tensile strain. In an environment below 50 °C, the loss of the fiber in U shape is less than 15% after repeated stretching 100 cycles in the strain range of 0~50%. The fiber has strong resistance to the side pressure, and can still pass about 30% of the light when subjected to 50 N side pressure. Compared with the straight fiber, the response sensitivity of the U-shaped fiber to deformation is increased by about 7 times at most. An additional communication-grade plastic optical fiber (attenuation less than 180 dB/km) is connected to the sensor as a reference signal. The reference fiber can effectively reduce the error of the sensor system. In addition, the sensors are easy to install so that the data gloves can be customized for different hand shapes. In the gesture capture test, the data glove can accurately identify and track the change of hand gestures in real time, and then guide the manipulator to make the correct action. The data glove is structurally stable, easy to manufacture and customizable. It can quickly, accurately and efficiently capture finger joint motion. It has broad development potential in motion monitoring, telemedicine and human-computer interaction. We believe that with the continuous development of virtual reality interactive technology, the application area of data glove will be further expanded.

## Figures and Tables

**Figure 1 polymers-15-00100-f001:**
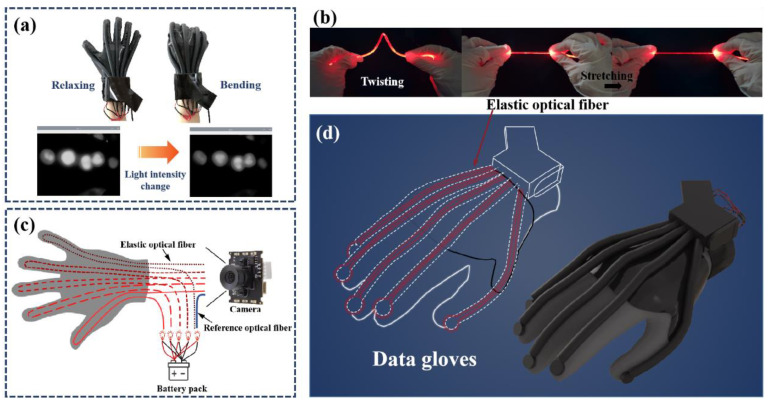
(**a**) Comparison of optical source before and after being affected by factors such as current. (**b**) Deformation display of the fiber. (**c**) The structure of the self-compensation module. (**d**) Structure diagram of data glove.

**Figure 2 polymers-15-00100-f002:**
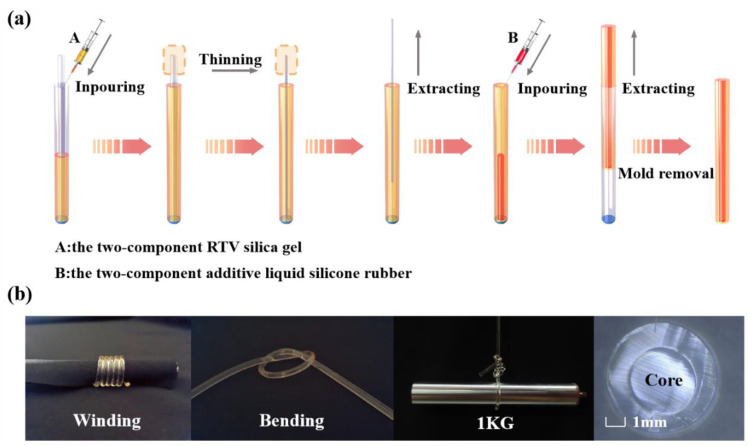
(**a**) Manufacturing process of elastic fiber. (**b**) The state of elastic fiber under winding and bending. And the cross-section of elastic fiber.

**Figure 3 polymers-15-00100-f003:**
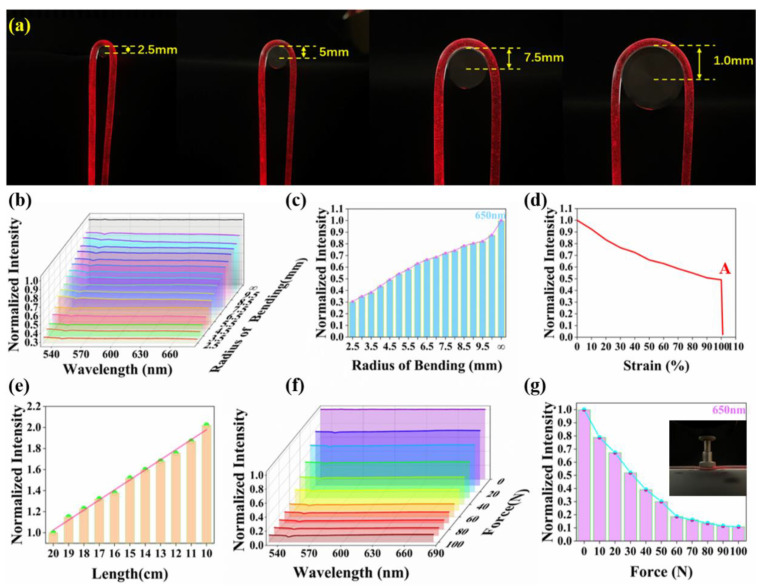
Mechanical and optical properties of elastic fibers. (**a**) Physical diagrams of U-shaped optical fiber with different bending radii. (**b**) Optical transmission changes of optical fibers under different bending radii. (**c**) The bending radius-normalized light intensity relation diagram at 650 nm. (**d**) Strain-normalized intensity curve of elastic fiber at 650 nm. (**e**) Propagation loss of elastic fiber, measured in air by cutback method. (**f**) The curves of optical transmission of the fiber subjected to a lateral pressure of 0 to 100 N. (**g**) The pressure-normalized light intensity relationship diagram at 650 nm.

**Figure 4 polymers-15-00100-f004:**
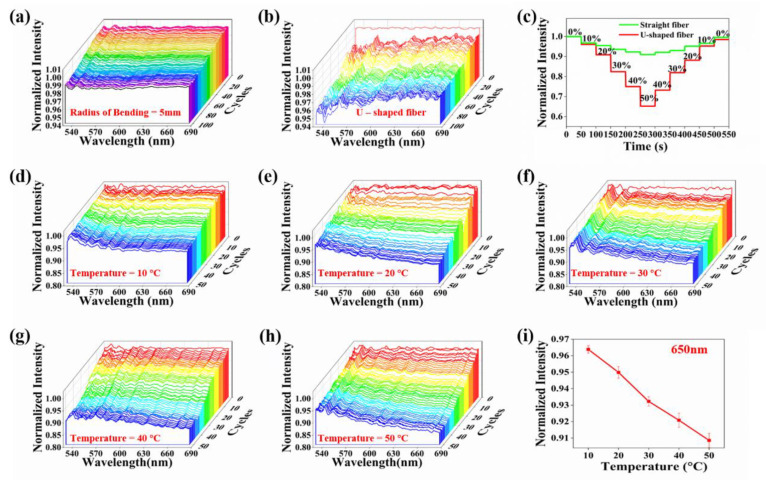
Bending and stretching sensing performance of the elastic fiber. (**a**) The elastic optical fiber is bent 100 cycles with a bending radius of 5 mm. (**b**) Tensile test of the U-shaped elastic fiber being stretched 100 cycles at the strain range 0~50%. (**c**) Comparison of sensing performance between straight and U-shaped elastic fibers during strain increase from 0 to 50% and back to 0. Tensile test of elastic fiber being stretched 50 cycles with 50% applied peak strain at different temperatures, (**d**) 10 °C, (**e**) 20 °C, (f) 30 °C, (**g**) 40 °C, (**h**) 50 °C. (**i**) The average normalized light intensity curve at 650 nm for five temperatures.

**Figure 5 polymers-15-00100-f005:**
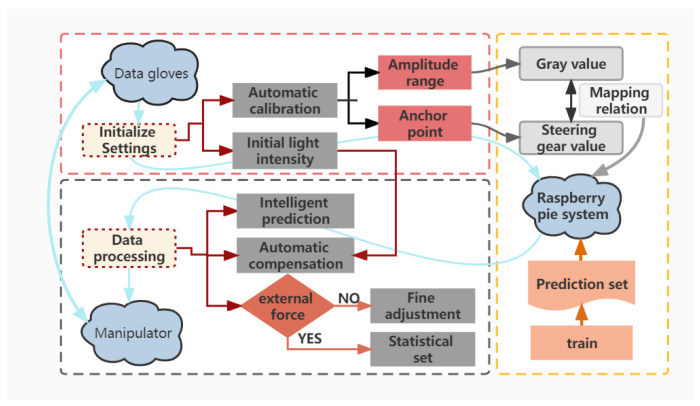
Flow chart of data collection and processing, including automatic calibration, intelligent prediction, automatic compensation and robust fine-tuning.

**Figure 6 polymers-15-00100-f006:**
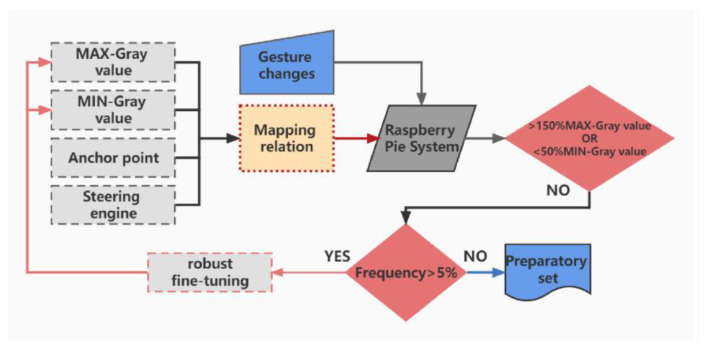
Flow chart of the robustness fine-tuning module.

**Figure 7 polymers-15-00100-f007:**
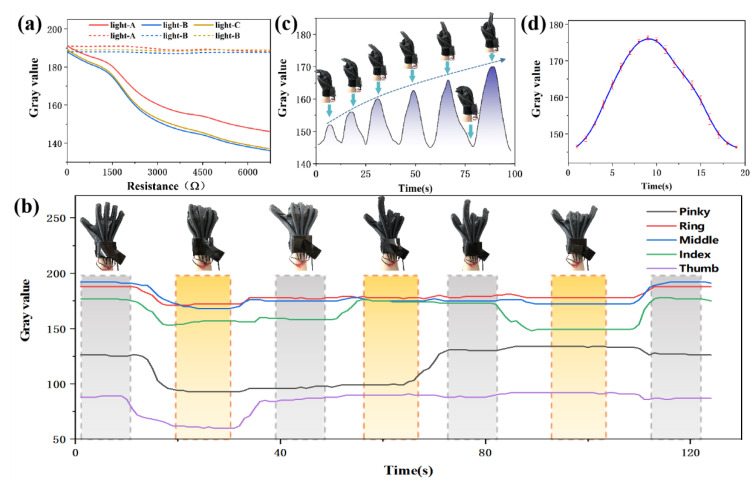
(**a**) Comparison between the brightness of the optical source without reference fiber and the brightness of the optical source after self-compensation processing during use. (**b**) Gray value curves of elastic fibers of five fingers under six consecutive different gestures. (**c**) The index finger is bent 6 times of different amplitude, and the bending radius is gradually increased until it is completely straight. (**d**) The average gray value curve of the index finger in the process of repeatedly bending and stretching 50 cycles.

**Figure 8 polymers-15-00100-f008:**
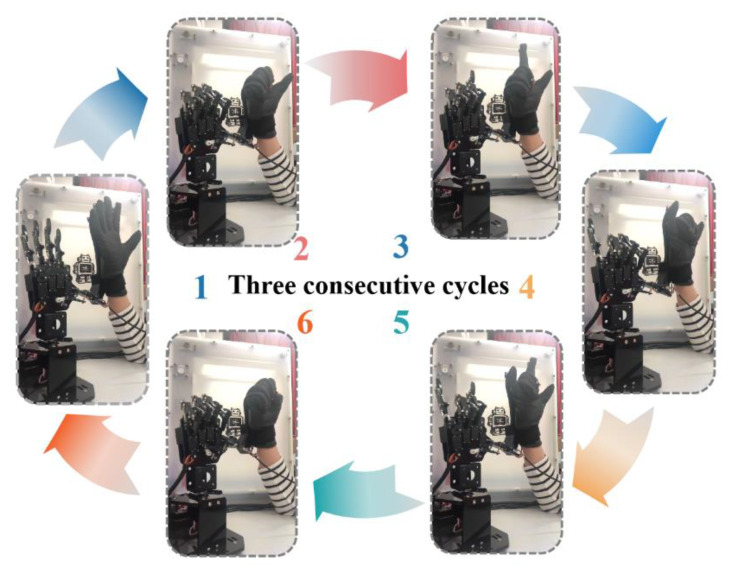
Video screenshots of Appendix A. The manipulator tracks the gestures of the data glove in real-time.

## Data Availability

Not applicable.

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
