# Peer review of "Data Glove with Self-Compensation Mechanism Based on High-Sensitive Elastic Fiber-Optic Sensor"

_polymers, 2022, doi:10.3390/polym15010100_

Round 1
Reviewer 1 Report
The authors proposed a low-cost smart glove based on self-compensating elastic optical fiber. The study is interesting. Nevertheless, the following questions should be answered and added to the manuscript to improve its clarity. Please check the following comments below:
1. The introduction should be rewritten to clarify the study's purpose.
2. The author should make figures describing the measurements mentioned in Figures 3 and 4 to clear the research results.
3. What are "syringes A and B"? Please provide more information such as manufacturer, datasheet, .etc.
4. The author should describe the intelligent prediction algorithm mentioned in section 3.2.
Reviewer 2 Report
Data Glove with Self-compensation Mechanism Based on High-sensitive Elastic Fiber-optic Sensor
In this manuscript, the main contribution of the authors is the development of a low-cost data glove based on a self-compensating elastic optical fiber sensor with self-calibration function for gesture capture.
The tunable and stretchable elastic fiber was fabricated by a simple, economical, and controllable method. It can not only maintain outstanding sensing properties at 10 °C to 50 °C, but also exhibit high stability after deformation such as stretching, bending and indentation. Further, a grade plastic fiber, POF is connected to the sensor as a reference signal, which can effectively reduce the error of the sensor system. The optical fibers are installed in the sensor in a U shape with a bending radius of 5mm. Compared with the straight fiber, the response sensitivity of the U-shaped fiber to deformation is substantially improved.
The proposal presented has broad development potential in motion monitoring, telemedicine and human-computer interaction; mainly due to the improvement of the device sensitivity with the incorporation of the low-cost U-shaped polymeric sensor and the incorporation of a reference POF to minimize errors in measurements.
I consider this paper is appropriate to be published in Polymers Journal after minor revised. Nevertheless, I would ask that the authors address the following matter:
It is suggested to improve the presentation of some figures such as Figs. 3, and 4. Particularly, the axes and the name of the axes, these are not clearly appreciated in the figures.
Reviewer 3 Report
This paper details the development and validation of a dataglove for recognition of hand gestures (to accompany VR applications). They have applied some AI techniques to recognise and allow for (natural) movement in neighbouring fingers e.g. when bending the middle finger. The authors have not tested the accuracy of the glove in measuring precise angles of flexion/extension as their main concern is to detect major changes of finger position.
The authors have carried out a number of tests to establish a fair degree of accuracy and repeatability under different conditions, including temperatures between 10 and 50 degrees C. They have applied a correction to account for loss of light as the battery fades. They have used a robotic hand to confirm accurate 'reading' of gestures from the glove. They suggest that lateral forces may cause inaccuracies but imply that this should not be a major problem.
The article is well written and well illustrated.
The validation testing appears to be based on one sample and repeat testing of the elastic properties were only carried out over a short period. It is not clear if the optical/tensile/elastic properties remain stable across manufacturing runs and over longer periods of time. The authors imply that this technology will have widespread applicability - but there are no details regarding cost/availability. I believe that this paper will be of interest to researchers in this field.
Round 2
Reviewer 1 Report
I am satisfied with the answers and additional explanations of the authors in the revised version of this manuscript and I suggest it be accepted.
